# Preparation of FeOOH/Cu with High Catalytic Activity for Degradation of Organic Dyes

**DOI:** 10.3390/ma12030338

**Published:** 2019-01-22

**Authors:** Yingzhe Zhang, Junfeng Liu, Ding Chen, Qingdong Qin, Yujiao Wu, Fang Huang, Wei Li

**Affiliations:** 1College of materials and metallurgical engineering, Guizhou Institute of Technology, Guiyang 550003, China; ljf12160610@163.com (J.L.); qin8370@126.com (Q.Q.); aluminum999@126.com (Y.W.); huangfangsqze@163.com (F.H.); 2Key Laboratory of Light Metal Materials Processing Technology of Guizhou Province, Guizhou Institute of Technology, Guiyang 550003, China; 32011 Collaborative Innovation Center of Guizhou Province, Guiyang 550003, China; 4State Key Laboratory of Advanced Design and Manufacturing for Vehicle Body, College of Mechanical and Vehicle Engineering, Hunan University, Changsha 410082, China; chending@hnu.edu.cn; 5School of Energy and Power Engineering, Changsha University of Science & Technology, Changsha 410014, China

**Keywords:** mechanochemistry, high-frequency electromagnetic-assisted ball-milling, FeOOH, catalysis

## Abstract

In this study, high-frequency electromagnetic-assisted ball-milling was used to prepare FeOOH/Cu catalyst. The combined effect of the high-frequency electromagnetic field and ball-milling resulted in the complete conversion of raw materials into FeOOH/Cu nanomagnetic hybrid at ~40 °C in only 30 h. Experiments showed that Rhodamine B was completely degraded within only 3 min, which was much faster than with previously reported catalysts. The combination effect of ball milling and microwave afforded excellent catalytic activity. Furthermore, the produced catalyst could be recovered easily using an external magnetic field for reuse. The influence of pH on the catalytic activity for degrading Rhodamine B, Phenol Red, Methyl Orange, and Methylene Blue were also investigated; Rhodamine B was completely degraded at pH 9 within only 2 min.

## 1. Introduction

Brightly colored dyes such as Rhodamine B, Phenol Red, Methyl Orange, and Methylene Blue are often released into rivers by the textile, printing, painting, and photography industries, thereby polluting the environment. These pollutants may poison aquatic animals and cause aquatic plant deaths by blocking the sun. Therefore, governments have severely restricted the discharge of organic colored dyes [1] and require industries to degrade them before discharge.

Catalysis is commonly used for degrading organic substances, and studies have developed various catalysts with excellent catalytic activity [2,3,4]. For example, Ramesh et al. developed α-MnO_2_ nanowires through a hydrothermal method; they showed excellent catalytic activity and degraded 70% of azo dye reactive black 5 in 60 min [5]. Tian prepared AC/TiO_2_ composites that could remove ~98% of Rhodamine B dye under ultraviolet (UV) irradiation for 30 min [6]. FeOOH is an ecofriendly oxide phase that usually acts as a natural regulator of nutrients or pollutants [7]. Recently, FeOOH and its compounds were demonstrated to be efficient catalysts [8,9,10]; they show good application prospects owing to their nontoxicity, good stability, and low cost. In this study, an FeOOH/Cu nanomagnetic hybrid was synthesized by a novel high-frequency electromagnetic-assisted ball-milling (HEABM) method, and its catalytic properties were studied.

Microwaves produce efficient thermal and non-thermal effects [11,12,13], and they can also effectively improve the catalytic activity of many catalysts [14]. Cai et al. experimentally demonstrated that, in a microwave field, the degradation efficiency of CuFeO_2_ for azo dye orange increased to 99.9% within only 15 min [15]. Lei et al. found that the hot spot effect of microwaves excited electron–hole and Fe^2+^–H_2_O_2_ reactions, and NiFeMnO_4_ showed degradation efficiency of up to 96.5% for 30.0 mg/L of Methyl Orange within 6 min [16].

## 2. Experimental

### 2.1. Preparation and Characterization

FeOOH/Cu catalyst was produced using the novel HEABM method [17] by the following process: CuO and Fe at a molar ratio of 1:1 were first put into a milling tank. Then, the milling balls and 300 mL of still water were added. The balls:reactant mass ratio was 50:1. Finally, the milling tank was put into a high-frequency electromagnetic field. The electromagnetic frequency was 200–300 KHz. The milling motor was turned on; the rotation speed of the stirrer was 258 r/min. The solution temperature in the HEABM process was maintained at ~40 °C throughout.

At a certain time point, the milled solution was taken out, filtered, and then dried at 323 K for 12 h. The evolution of transformation products was monitored using X-ray diffraction (XRD, dan dong tong da, dan dong, china). The particle sizes and shapes were imaged by high-resolution transmission electron microscopy (HRTEM, JEOL, Tokyo, Japan) and selected area electron diffraction (SAED, JEOL, Tokyo, Japan).

### 2.2. Microwave Catalytic Processes

The microwave catalytic activity was evaluated by the microwave-induced degradation of Rhodamine B, Phenol Red, Methyl Orange, and Methylene Blue. Microwave irradiation was conducted in a microwave oven with a rated power of 700 W and a frequency of 2450 MHz. First, 0.275 mg of the FeOOH/Cu catalyst and 10 mL of H_2_O_2_ were added to 60 mL of the dye solution (54 mg/L). The solution was placed in the dark for 12 h so that it would reach absorption–desorption equilibrium. Then, it was placed in a microwave oven. In the catalytic degradation experiment, microwave irradiation for 15 s was followed by a 15 s interval.

To illustrate the special effect of microwaves in the catalytic experiment, a regular hot water-assisted catalytic experiment was designed for comparison purposes. In the comparison experiment, no microwave was used and 60 ml of Rhodamine B solution (54 mg/L) with 10 mL of H_2_O_2_ and 0.275 mg of FeOOH/Cu catalyst were placed in the dark for 12 h and then heated in hot water.

During all of the above-described catalytic degradation experiments, the organic pigment content was detected using a JINGHUA (Shanghai, China) UV–Vis 756 spectrophotometer.

## 3. Results and Discussion

### 3.1. Analysis of Progress

The phase evolution was detected by X-ray diffraction (XRD), and the phases appearing in the HEABM method were characterized as shown in Figure 1. The patterns indicate that all Fe phases disappeared and a lot of FeO emerged along with a little Cu in the first 6 h. With an increased milling time, the amount of Cu increased and the amount of CuO decreased. After 18 h, FeOOH appeared. When the experiment was continued for 30 h, only the peaks belonging to FeOOH and Cu remained, indicating complete conversion of the raw materials into FeOOH and Cu. These results indicate that HEABM effectively reduces the reaction temperature, enhances the product yield, and shortens the reaction time.

Furthermore, XRD patterns showed all intermediate and final products had no residual pollutants that would be discharged into the environment. Therefore, HEABM is an ecofriendly method for producing nanopowders.

Based on the XRD results and discussion above, the following chemical reactions occur in HEABM:
CuO + Fe → FeO + Cu(1)
2FeO + O_2_ + H_2_O → 2FeOOH.(2)

Ball milling was used to break the CuO particles and large Fe phase into smaller particles. This process stored considerable stress, strain, and dislocation in the smaller particles [18] and thus enhanced their activity energy. At the same time, the smaller particles had higher plasticity, could bond together easily [19], and showed reduced diffusion distance.

In the experiment, the combination of the high-frequency electromagnetic field and ball milling played a key role. The former causes electrons on the Fe surface to move quickly. The latter causes CuO and Fe to combine, whereby electrons and Cu^2+^ make contact and reaction (1) occurs. Furthermore, the solution was stirred by ball milling, and O_2_ can be added easily to the solution to oxidize FeO into FeOOH and cause reaction (2) to occur.

### 3.2. Analysis of Morphology

Figure 2a shows the morphology of the FeOOH/Cu particles prepared by HEABM in 30 h. The pattern indicates that the produced particles are well-dispersed and that most have ~20 nm size. Solid phases broke into smaller particles via ball milling to form nanoparticles. Electrons distributing in the milled irregular nanoparticles were usually uneven and dipoles formed [20]. In the HEABM process, dipoles tend to rotate such that their directions coincide with the external electric field [21]. As they rotate, they collide and rub against each other, and therefore, corners wear continuously. After some time, the particles become nearly spherical. Figure 2b shows that all bright bots correspond to FeOOH and Cu. This suggests that the raw materials are converted completely; this agrees with the XRD results. Furthermore, the diffraction pattern consists of many bright diffraction spots but no diffraction ring, indicating that the particles make up a well-ordered lattice. Figure 2c shows the morphology of the produced catalyst; Cu and FeOOH particles are seen to be well connected. When an external magnetic field is applied, both Cu and magnetic FeOOH particles in the aqueous solution can be recovered together quickly for reuse, as shown in Figure 2d.

### 3.3. Analysis of Catalytic Performance

Microwave-induced catalytic degradation shows promise for purifying wastewater [6,22,23]. Figure 3a shows the UV–vis spectrum of the solution when irradiated by microwaves for different durations. When the solution was irradiated for 0.5 min, four absorption peaks corresponding to rhodamine B were seen at ~263.3, 355.6, 549.4, and 561.1 nm. The strengths of all peaks decreased as the irradiating time increased; after 3 min, they disappeared completely, indicating that the dye was degraded completely. The color of the rhodamine B aqueous solution became fainter as the degradation time increased, and the solution became clear and transparent when it was irradiated for 3 min.

A microwave field is a combination of an electric field and a magnetic field whose orientation is changed at a high frequency of 2450 MHz. Thus far, several mechanisms have been proposed to explain the synergistic effect between microwaves and a catalyst. Generally, the combination of microwaves with microwave-absorbing materials can produce a large number of hot spots on the surfaces of these materials [24,25]. More rapid oxidation and combustion of organic pollutant molecules could occur, thus leading to high degradation efficiency. In this study, the enhanced catalytic property may have been induced by the special structure of Cu_2_O. Figure 2a shows that the synthesized Cu_2_O was nanosized and could absorb the dyes easily. By contrast, the combination of microwave and ball milling plays a key role in the microwave-assisted catalytic experiment. Owing to ball milling in the process of preparing FeOOH/Cu by HEABM, electrons in the catalyst are mostly distributed unevenly, thereby inducing a dipole moment. Then, in the subsequent microwave-induced catalytic degradation experiment, these dipole moments align themselves with the oscillating electric field of the microwave irradiation, leading to high-frequency shaking and rubbing against each other. In this situation, the catalytic effect of the synthesized FeOOH/Cu in the microwave field is very high.

Figure 3b shows the UV–vis spectrum of the solution in hot water heated by the regular method. The patterns indicate that the color remained red for 30 min. Most of the absorption peaks for rhodamine B, except for the one at 549.4 nm, disappeared, indicating that a little rhodamine B was still left in the solution.

To investigate the recyclability and stability of the FeOOH/Cu catalyst, we recycled the catalysts from the solution using a magnet. Then, microwave-induced catalytic degradation experiments using the recycled catalyst were repeated 10 times; the obtained results are shown in Figure 3c. The pattern indicates that the time required for degrading rhodamine B completely increased a little with the recycling time. This may have been induced by the deactivation of the catalyst. The XRD pattern of the catalyst used 10 times is shown in Figure 3d. The figure indicates little difference from the pattern of the original catalyst reused 10 times. Although the structure of the catalyst is stable and it can be recovered for reuse for many times, the catalytic ability decreased as the reusing time increased. This means that catalyst structure is not an all-activity material.

For a better comparison, the catalytic ability of FeOOH and its compounds as synthesized by different approaches previously and in this study are listed in Table 1. The as-prepared FeOOH/Cu powder clearly showed higher catalytic efficiency for degrading rhodamine B than previously reported catalysts. The proposed HEABM method is therefore effective for preparing FeOOH/Cu with excellent catalytic activity.

The influence of pH on the degradation of Rhodamine B was also investigated. Figure 4a shows a standard UV–vis absorption pattern at 549.4 nm for Rhodamine B solutions of different percentages. The percentage of Rhodamine B in the solution was calculated as
Rhodamine B percent (%) = (C_x_/C_0_) × 100%(3)
where C_0_ is the initial concentration of Rhodamine B and C_x_ is the concentration of Rhodamine B prepared initially.

The intensity of the UV–vis absorption peak at 549.4 nm for Rhodamine B solutions of different percentages was detected and a curve of intensity versus Rhodamine B percentage was plotted, as shown in Figure 4a. The percentage of Rhodamine B was proportional to the intensity up to 50%. For higher percentages, the proportional relationship no longer existed, and therefore, the percentage was directly evaluated from the curve.

To study the influence of pH on the degradation of Rhodamine B, HCl and NaOH were added to the Rhodamine B solution; the resulting pattern is shown in Figure 4b. The pattern indicates that the degradation efficiency of Rhodamine B increased up to a pH of 9. This suggests that OH^-^ enhanced the degradation of Rhodamine B. In particular, Rhodamine B was degraded completely in only 2 min at pH 9.

Usually, wastewater from printing and painting industries contains organic dyes. Therefore, it is meaningful to develop a catalyst that can degrade various organic pollutants. Figure 5 shows the UV–vis diffuse reflection spectra of Phenol Red solution, Methyl Orange solution, and Methylene Blue solution irradiated by microwaves for different durations and the corresponding changes in color. The patterns indicate that the reflection peaks corresponding to Phenol Red solution and Methylene Blue solution disappeared within only 2 min, and the colored water correspondingly became colorless and transparent. Furthermore, Methyl Orange was degraded completely within 3 min and the red color faded. These results suggest that the prepared FeOOH/Cu showed excellent catalytic activity.

## 4. Conclusions

In this study, FeOOH/Cu nanocatalyst was synthesized using a novel HEABM method in only one step without any sintering process. The solution temperature was maintained below 40 °C throughout. No organic solvents were used, and no waste gas or residue was produced in this process, and it was confirmed that HEABM is an ecofriendly method with high efficiency for product preparation, a short preparation time, and a high conversion rate of raw materials.

The prepared FeOOH/Cu showed much higher catalytic ability for degrading Rhodamine B than previously reported catalysts. This catalyst could completely degrade dyes such as Rhodamine B, Phenol Red, Methyl Orange, and Methylene Blue within 3 min. Furthermore, at pH 9, Rhodamine B was completely degraded within 2 min. This catalyst showed excellent catalytic ability, making it promising for applications involving the degradation of environmental pollutants.

## Figures and Tables

**Figure 1 materials-12-00338-f001:**
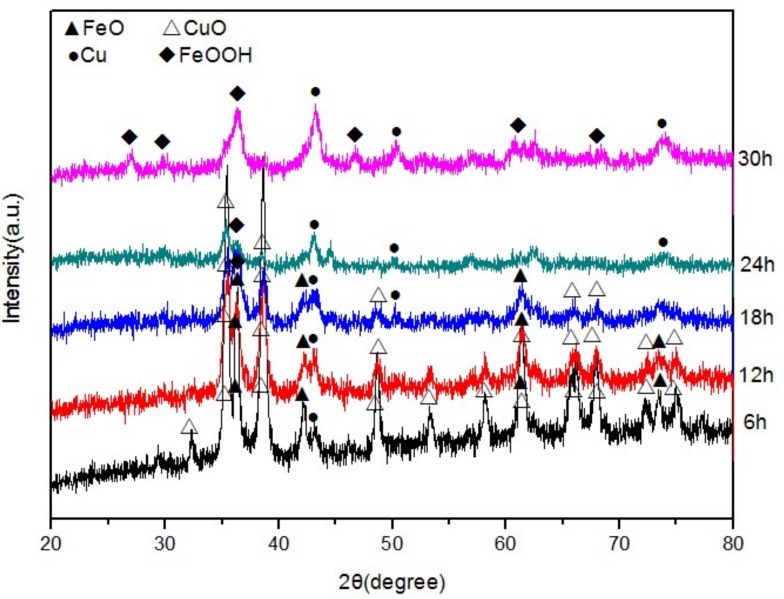
X-ray diffraction (XRD) patterns of products appearing in the process at different times.

**Figure 2 materials-12-00338-f002:**
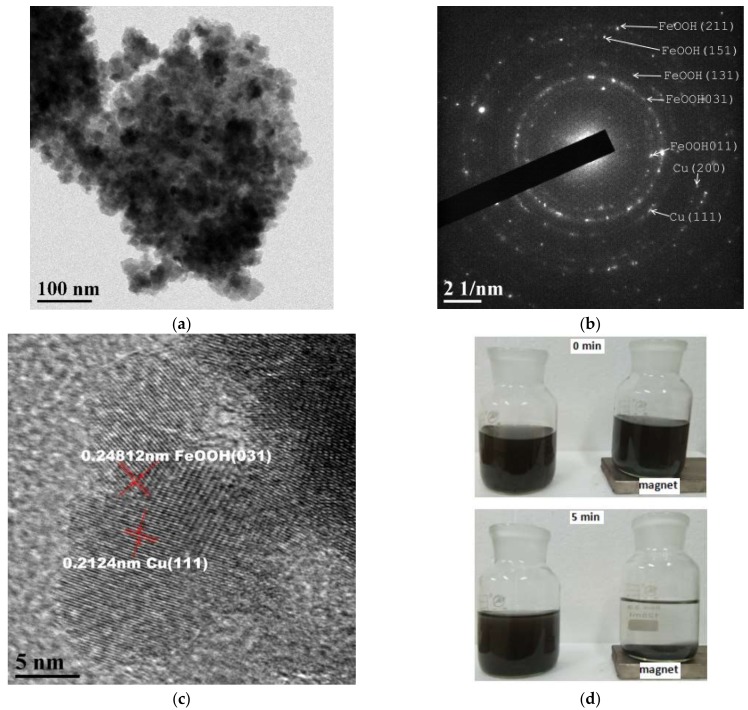
(**a**) Microstructure, (**b**) selected area electron diffraction (SAED), (**c**) high-resolution transmission electron microscopy (HRTEM), and (**d**) macroscopic morphology of FeOOH/Cu in aqueous solution.

**Figure 3 materials-12-00338-f003:**
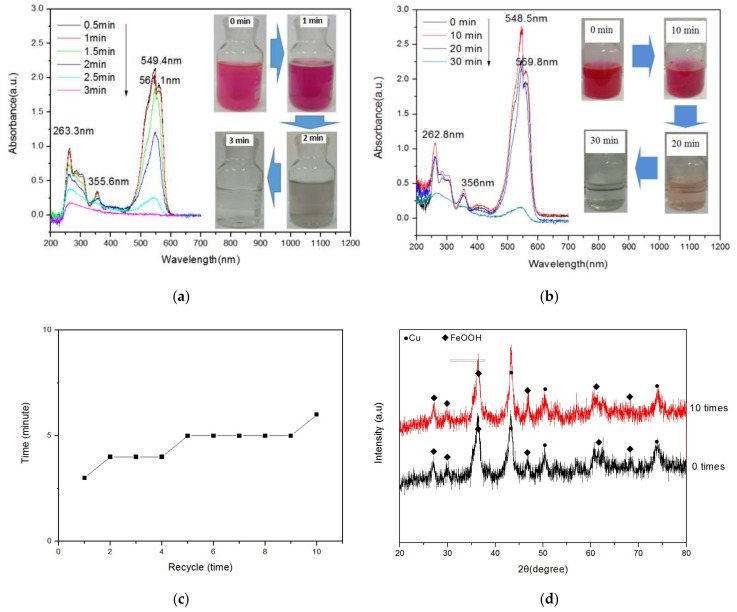
(**a**) UV–vis diffuse reflection spectra of Rhodamine B solution degraded in a microwave field for different durations and corresponding color change. (**b**) UV–vis diffuse reflection spectra of Rhodamine B solution degraded in hot water for different durations and corresponding color changes. (**c**) The time required for degrading Rhodamine B completely with recycling times. (**d**) XRD patterns of original FeOOH/Cu that were reused 10 times.

**Figure 4 materials-12-00338-f004:**
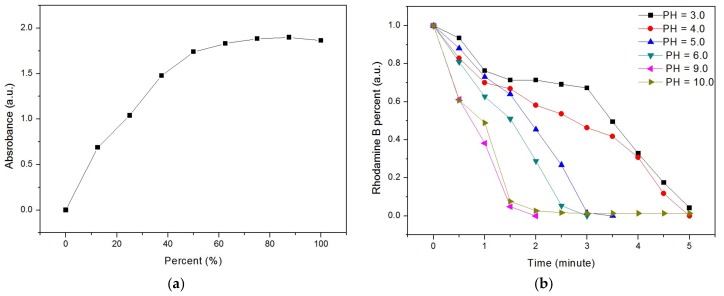
(**a**) Standard absorption pattern at 549.4 nm for Rhodamine B solution of different percentages. (**b**) Degradation of Rhodamine B at different pH values.

**Figure 5 materials-12-00338-f005:**
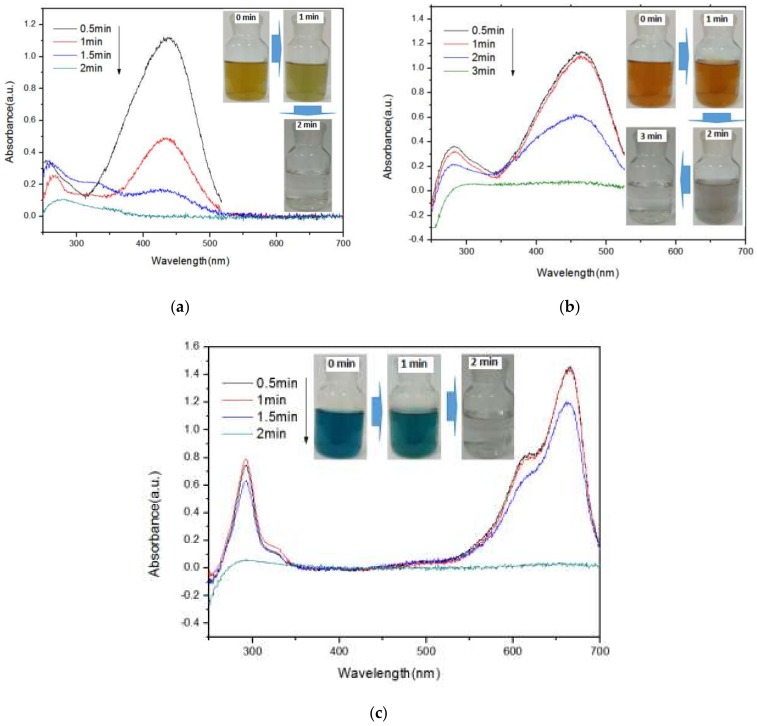
(**a**) UV–vis diffuse reflection spectra of Phenol Red solution, (**b**) Methyl Orange solution, and (**c**) Methylene Blue solution degraded for different durations and corresponding change in color.

**Table 1 materials-12-00338-t001:** Catalytic activity of FeOOH synthesized by different approaches for degrading Rhodamine B.

Synthesis Technology	Catalyst	Rhodamine B Degradation Rate	Time (min)	References
Co-precipitation	FeOOH	87%	60	[26]
Hydrothermal	FeOOH/WO_3_·H2O	12%	210	[27]
Low-temperature water bath	FeOOH	100%	60	[28]
Assembly method	FeOOH	100%	70	[8]
Glucose assisted hydrothermal	BiOCl/β-FeOOH	100%	60	[29]
HEABM	FeOOH/Cu	About 100%	3	Present study (in microwave field)
HEABM	FeOOH/Cu	Almost 95%	30	Present study (in hot water)

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
