# Peer review of "Preparation of FeOOH/Cu with High Catalytic Activity for Degradation of Organic Dyes"

_materials, 2019, doi:10.3390/ma12030338_

Reviewer 1 Report

This manuscript reports the preparation and catalytic assessment of an Fe/Cu catalyst for degradation of dyes. The work is interesting and fits the aims & scope of the journal. However, there are some points that must be revised:

1. In section 2.2, authors should benchmark the microwave catalytic activity against that from regular chemistry without the microwave radiation to effectively assess that effect.

2. In line 59, authors mention “Methyl blue”; this should be “Methylene blue” and needs correction.

3. In Figure 3, the spectra of Rhodamine B look very strange. It seems saturated.

4. Why have authors measured diffuse reflectance UV/Vis spectra of the dyes? These form clear solutions and their spectra could be measured in solution using regular transmission setup.

5. In Figure 5, How were these experiments conducted?

6. The catalysts have the same structure and morphology after the catalytic experiments? Add the characterization (XRD, SEM/TEM, etc…) and discuss it after each catalytic experiment.

7. Is it possible to recycle the catalyst? Do some experiments and add to the manuscript.

As a final comment, the English spelling needs a very strong revision.

Therefore, I recommend major revision of this manuscript.

Author Response

Many thanks for the comments from you on our paper “Preparation of FeOOH/Cu with high catalytic activity for degradation of organic dyes” submitted to “materials”. We have checked the manuscript carefully and revised it extensively according to the comments. And in the revised manuscript,the changes of mine have been highlighted using a red color so that they can easily be identified.

Q1. In section 2.2, authors should benchmark the microwave catalytic activity against that from regular chemistry without the microwave radiation to effectively assess that effect.

R1. In order to research the special effect of microwave in the catalytic experiment, a comparing experiment of regular hot water assisted catalytic experiment was designed and the results are shown as figure 3(b). please see line 73-77, 145-150 and figure 3(b) in the revised manuscript..

Q2. In line 59, authors mention “Methyl blue”; this should be “Methylene blue” and needs correction.

R2. The “Methyl blue” in the manuscript has been replaced by “Methylene blue”. Please see  line 67 in the revised manuscript.

Q3. In Figure 3, the spectra of Rhodamine B look very strange. It seems saturated.

R3. The spectra in figure 3 is an original diagram and it was not saturated.

Q4. Why have authors measured diffuse reflectance UV/Vis spectra of the dyes? These form clear solutions and their spectra could be measured in solution using regular transmission setup.

R4. Diffuse reflectance UV/Vis spectra of the dyes have been measured in some reported articles and in order to research the related property of the synthesized FeOOH/Cu, we measured the diffuse reflectance UV/Vis spectra of the dyes. The following are the reported articles: [Yang G, Jiang Z, Shi H, et al. Preparation of highly visible-light active N-doped TiO 2 photocatalyst[J]. Journal of Materials Chemistry, 2010, 20(25): 5301-5309.],[Miao S, Liu Z, Han B, et al. Synthesis and characterization of TiO 2–montmorillonite nanocomposites and their application for removal of methylene blue[J]. Journal of Materials chemistry, 2006, 16(6): 579-584.] and [Zhang M, An T, Hu X, et al. Preparation and photocatalytic properties of a nanometer ZnO–SnO2 coupled oxide[J]. Applied Catalysis A: General, 2004, 260(2): 215-222.].

Q5. In Figure 5, How were these experiments conducted?

R5. The microwave-catalytic activity was evaluated by the microwave-induced degradation of Rhodamine B, Phenol Red, Methyl orange, and Methylene blue. Microwave-irradiation was conducted in a microwave oven with rated power of 700 W and frequency of 2450 MHz. First, 0.275 mg of the FeOOH/Cu catalyst and 10 mL of H2O2 were added to 60 mL of dye solution (54 mg/L). The solution was placed in the dark for 12 h so that it would reach absorption-desorption equilibrium. Then, it was placed in a microwave oven. In the catalytic degradation experiment, microwave irradiation for 15 s was followed by a 15-s interval. Please see line 66 to line 72 in the revised manuscript.

Q6. The catalysts have the same structure and morphology after the catalytic experiments? Add the characterization (XRD, SEM/TEM, etc…) and discuss it after each catalytic experiment.

R6. X-ray diffraction technology was used to characterize the catalyst reused for 10 times and the result was showed as figure 3(d). From the figure it can been seen that there is not much different form the pattern of original catalyst to that reused for 10 times. It means that the prepared FeOOH/Cu is stable for repeated reuse in the future. Please see line 157 to line 160 in the revised manuscript.

Q7. Is it possible to recycle the catalyst? Do some experiments and add to the manuscript.

R7. In order to investigate the recycle ability and stability of the synthesized FeOOH/Cu catalyst, FeOOH/Cu in the solution was recycled by a magnet. Then the microwave-induced catalytic degradation experiment using the recycled catalyst was repeated for 10 times and the result was showed as figure 3(c). From the pattern we can see that the time for degrading the Rhodamine B completely increased as the recycle times increased. It may be due to that some changes happened happened to the catalyst in the catalytic experiment. However, the time used for degrading rhodamine B by using the prepared catalyst was also much short. Further more, X-ray diffraction technology was used to characterize the catalyst reused for 10 times and the result was showed as figure 3(d). From the figure it can been seen that there is not much different form the pattern of original catalyst to that reused for 10 times. It means that the prepared FeOOH/Cu is stable for repeated reuse in the future. Please see line 151 to line 160 in the revised manuscript.

Finally, we really appreciate your hard work, instructive advice and kindly help to our paper. If you have any questions about the revised manuscript, please let us know.

Reviewer 2 Report

The paper provides a description of thorough investigation concerning the preparation of FeOOH/Cu catalyst with high catalytic activity for degradation of organic dyes. High-frequency electromagnetic-assisted ball-milling was used to prepare the catalyst. Presented results are very interesting but a more detailed explanation and elaboration of the topic is necessary to propose and understand the catalytic reaction mechanism. This quite interesting paper is probably publishable, however, I am afraid that the work in the present quite short form would appeal only to a limited group of researchers. This is due to the fact that the Authors only inform about the obtained results concerning catalytic activity of the catalyst studied. Simple conclusion about that may of course be interesting (but it is not enough for this paper to be published). However, there is something missing, namely there is no conclusion about the reaction mechanism in the paper what, in my opinion, is even more interesting than the catalytic activity itself. This short form can be justified for ground-

breaking papers but I have the impression that this is not the case, since I am not convinced about the novelty of the work (similar systems were investigated what was also stated by the Authors). Due to the short form of the manuscript I suggest to publish it as a letter if any. Additionally, the work is worth publishing, however, I am not convinced whether there will be a broad audience interested in this paper since it seems to be rather intended for a quite specialized group of researchers. To sum up, this quite interesting paper is probably publishable, but (in my opinion) only after major revision to add the information about the reaction model on FeOOH/Cu catalyst studied. The novelty and ground-breaking character of the work is also not convincing as for now.

Author Response

Many thanks for the comments from you on our paper “Preparation of FeOOH/Cu with high catalytic activity for degradation of organic dyes” submitted to “materials”. We have checked the manuscript carefully and revised it extensively according to the comments. And in the revised manuscript,the changes of mine have been highlighted using a red color so that they can easily be identified.

In my opinion, reaction mechanism in the microwave induced catalytic experiment is as following: Microwave field is the combination of electric field and magnetic field changing their orientation at a high frequency of 2450 MHz. Until now, several mechanisms were proposed to explain the synergistic effect between microwave and catalyst. Generally, it was accepted that the combination of microwave with microwave-absorbing materials can produce great amounts of “hot spots” on the surface of these materials, on which a more rapid oxidation and combustion of organic pollutant molecules could occur, thus leading to high degradation efficiency. In this manuscript, the enhanced catalytic property may be induced by the special structure of FeOOH/Cu. From Figure 2 we can see that the synthesized FeOOH/Cu particles were in nano size so that they may absorb the dyes onto their surface easily. On the other hand, because of the function of ball milling in the process of preparing FeOOH/Cu by HEABM, the electrons in the catalyst are mostly distributed uneven and dipole moment induced. Then in the following microwave-induced catalytic degradation experiment, these dipole moment attempt to align themselves with the oscillating electric field of the microwave irradiation leading to shaking at a high frequency and rubbing against each other. In this situation, much more “hot spots” may be induced between these colliding particles, so that the prepared FeOOH/Cu showed excellent catalytic activity in the microwave catalytic degradation experiment.

Further more, In order to research the special effect of microwave in the catalytic experiment, a comparing experiment of regular hot water assisted catalytic experiment was designed. In order to investigate the recycle ability and stability of the synthesized FeOOH/Cu catalyst, FeOOH/Cu in the solution was recycled by a magnet. Then the microwave-induced catalytic degradation experiment using the recycled catalyst was repeated for 10 times and the result was showed as figure 3(c). From the pattern we can see that the time for degrading the Rhodamine B completely increased as the recycle times increased. It may be due to that some changes happened happened to the catalyst in the catalytic experiment. However, the time used for degrading rhodamine B by using the prepared catalyst was also much short. Further more, X-ray diffraction technology was used to characterize the catalyst reused for 10 times and the result was showed as figure 3(d). From the figure it can been seen that there is not much different form the pattern of original catalyst to that reused for 10 times. It means that the prepared FeOOH/Cu is stable for repeated reuse in the future.

Please see line 73-77, 132-160 and figure 3(b) , figure 3(c) and figure 3(d) in the revised manuscript..

Finally, we really appreciate your hard work, instructive advice and kindly help to our paper. If you have any questions about the revised manuscript, please let us know.

Round  2

Reviewer 1 Report

Accept in present form.

Author Response

We really appreciate your hard work and many thanks for the recognition from you on our paper “Preparation of FeOOH/Cu with high catalytic activity for degradation of organic dyes” submitted to “materials” .

Reviewer 2 Report

First of all, I would like to thank for the elaborated answer in the covering letter. For the purpose of that letter, it is fine to answer the way Authors did. I mean, they state their opinion and present some ideas ("In my opinion, reaction mechanism in the microwave induced catalytic experiment is as following" and so on). However, such a form is not satisfactory in the manuscript without scientific proofs, since for now this explanation is a question of Authors' 'beliefs' rather than 'hard evidences'. Whilst in the covering letter the explanation seems to be OK, more 'scientific' addition must be included in the manuscript: first of all citations on e.g. "several mechanisms were proposed", "it was accepted that the combination of microwave with microwave-absorbing materials can produce great amounts of “hot spots” on the surface of these materials," (who did performed previous experiments? what did they discover? what are the experimental evidences of the Authors that the idea about the reaction model is true? Please, complete the 'manuscript version' in more 'scientific' way.

Apart from that, there is some inconsistency in the Authors' answers about the stability of the catalyst. They state that the catalyst is stable, because the structure did not change. On the other hand, they observed that the activity change during longer period of time (Fig. 3c - ca. 100% longer time from 3 to 6 min). The answer should be that catalyst is not stable despite the stable structure, since structure it is not all - activity matters. Strange scale in Fig. 3c - why does it reach 30 min not e.g. 10 min?

Author Response

Many thanks for the comments from you on our paper “Preparation of FeOOH/Cu with high catalytic activity for degradation of organic dyes” submitted to “materials”. We have checked the manuscript carefully and revised it extensively according to the comments. And in the revised manuscript,the changes of mine have been highlighted using a red color so that they can easily be identified.

(1) English language and style have been re- edited and the changes in the manuscript have been highlighted using a red color.

(2) Thanks very much for your scientific and rigorous suggestion. Because of the high frequency (2450MHz) of the microwave the hot spots arise and disappear instantaneously. So that it is too difficult for sensor to captured it. However, by reviewing the reported literatures, we found that hot spot should exist in the process of Microwave-induced catalytic degradation experiments. Kai xi et al believe that the combination of dielectric loss and magnetic loss of the material contributed to the ability to produce more “hot spots”. These hot spots promoted the oxidation of common antibiotics [Shiyuan L , Lefu M , Xiaoliang L , et al. Anchoring Fe3O4 nanoparticles on carbon nanotube for microwave-induced catalytic degradation of antibiotics[J]. ACS Applied Materials & Interfaces, 2018:acsami.8b08280-.]. Yonglin Lei et al demonstrated that in the reported article the high removal efficiency of methyl orange for NiFeMnO4-MW-H2O2 process was attributed to joint reaction with the direct decomposition by MW “hot spots”, the MW “hot spots” accelerated electron–hole excitation and Fe2+-H2O2 reaction [Lei Y, Lin X, Liao H. New insights on microwave induced rapid degradation of methyl orange based on the joint reaction with acceleration effect between electron hopping and Fe2+-H2O2 reaction of NiFeMnO4 nanocomposites[J]. Separation and Purification Technology, 2018, 192: 220-229.].

In this manuscript, indirect evidence for the existence of the hot sports in this manuscript can been gotten by an comparing experiment of hot-water-assisted catalytic. Figure 3(b) shows the UV–vis spectrum of the solution in hot water heated by the regular method. The patterns indicate that the color remained red for 30 min. Most of the absorption peaks for Rhodamine B, except for the one at 549.4 nm, disappeared, indicating that a little Rhodamine B is still left in the solution. It may means that it is the hot spots that should play a major role in degrading Rhodamine B in the Microwave-induced catalytic degradation experiment.

Please see line 73-77, 136, 148-151 and figure 3(b) in the revised manuscript

(3) Thanks for your advise and the inconsistency in the manuscript have been changed as: Although the structure of the catalyst is stable and it can be recovered for reusing for many times, the catalytic ability decreased as the reusing time increased. It means that catalyst structure is not all - activity matters.   

Please see line 159-161 in the revised manuscript

(4) Furthermore, the scale in Fig. 3c has been changed to 10 min.

Finally, we really appreciate your hard work, instructive advice and kindly help to our paper. If you have any questions about the revised manuscript, please let us know.

 Round  3

Reviewer 2 Report

I would like to thank the Authors for the presented answers.